# Fascinating Molecular and Immune Escape Mechanisms in the Treatment of STIs (Syphilis, Gonorrhea, Chlamydia, and Herpes Simplex)

**DOI:** 10.3390/ijms23073550

**Published:** 2022-03-24

**Authors:** Lucian G. Scurtu, Viorel Jinga, Olga Simionescu

**Affiliations:** 1Department of Dermatology I, Colentina Clinical Hospital, Faculty of Medicine, Carol Davila University of Medicine and Pharmacy, 020125 Bucharest, Romania; lucian.scurtu@drd.umfcd.ro; 2Department of Urology, Clinical Hospital Prof. Dr. Th. Burghele, Faculty of Medicine, Carol Davila University of Medicine and Pharmacy, 030167 Bucharest, Romania; viorel.jinga@umfcd.ro

**Keywords:** syphilis, gonorrhea, chlamydia, herpes simplex, antimicrobial, nanoparticles, imiquimod

## Abstract

The incidence of syphilis, gonorrhea, chlamydia, and herpes simplex has increased over the last decade, despite the numerous prevention strategies. Worldwide scientists report a surge in drug-resistant infections, particularly in immunocompromised patients. Antigenic variations in syphilis enable long-term infection, but benzathine penicillin G maintains its efficiency, whereas macrolides should be recommended with caution. Mupirocin and zoliflodacin were recently introduced as therapies against ceftriaxone-resistant gonococcus, which poses a larger global threat. The gastrointestinal and prostatic potential reservoirs of Chlamydia trachomatis may represent the key towards complete eradication. Similar to syphilis, macrolides resistance has to be considered in genital chlamydiosis. Acyclovir-resistant HSV may respond to the novel helicase-primase inhibitors and topical imiquimod, particularly in HIV-positive patients. Novel drugs can overcome these challenges while nanocarriers enhance their potency, particularly in mucosal areas. This review summarizes the most recent and valuable discoveries regarding the immunopathogenic mechanisms of these sexually transmitted infections and discusses the challenges and opportunities of the novel molecules and nanomaterials.

## 1. Introduction

Sexually transmitted infections (STIs) are among the most common acute diseases worldwide. Despite the numerous local efforts and national prevention strategies, their incidence remains high. Long-term complications such as infertility, seronegative arthritis, and neurological disorders are worrisome [1]. Nevertheless, in this era of medicine focused on subspecialties, Venerology has not yet seen a surge of interest in publications. Venerology is usually assigned to dermatologists who focus their research on non-venereal pathology. The Centers for Disease Control and Prevention (CDC) estimated 2 million cases of Chlamydia, gonorrhea, and syphilis, every single year, in the United States only. Chlamydia represents the most prevalent bacterial STI in the United States, followed by gonorrhea. Of note, syphilis (“The Great Imitator”) prevalence has drastically increased in the United States lately [2]. An estimated half a billion people worldwide have herpes simplex virus (HSV) genital infection [3]. CDC identified drug-resistant *Neisseria gonorrhoeae* among the top five urgent antibiotic-resistant threats to public health, but the impact of antimicrobial-resistant (AMR) STIs is often underestimated. Furthermore, foreign travel contributes to the spread of AMR infections, as one-third of international travelers engage in casual travel intercourse [4].

In syphilis, *Treponema pallidum* rare outer membrane (OM) proteins (TROMPs) are the main antigens recognized by the host Toll-like receptors (TLRs) [5], but the antigenic variation of these surface lipoproteins promotes immune escape [6]. Chromosomal transfer of AMR genes is an important factor, that may lead to AMR gonorrhea [7]. The emergence of extensive drug-resistant (XDR) gonococcus strains in Japan, France, and Spain and the first resistant strain to ceftriaxone and azithromycin combined therapy in England and Australia [8] has sparked a global trend towards discovering novel therapeutic approaches in gonorrhea [9]. *Chlamydia trachomatis* is a leading bacterial STI in developed and undeveloped countries [10]. The alternation between two morphological different forms enables particular metabolic activities and confers resistance features to Chlamydia species [11].

Nucleoside analogs such as acyclovir have paved the way to HSV infection treatment [12], but the prevalence of acyclovir-resistant HSV exceeds 10% in solid organ and hematopoietic stem cell transplant recipients [13], and alternatives are warranted in this subset of patients, as well as in HIV positive patients [14]. This paper aims to describe the molecular and immunological mechanisms in syphilis, gonorrhea, chlamydia, and herpes simplex infections in relation to currently available treatments and drug resistance, respectively.

## 2. Host Interactions

### 2.1. Syphilis

*Treponema pallidum*, subsp. pallidum, phylum Spirochaetes, is a spiral-shaped, dark-field visible, extracellular bacterium that causes syphilis. Venereal syphilis is usually sexually transmitted through skin or mucosal microabrasions. After an incubation period varying from 9 days up to 2–3 months, an asymptomatic, highly contagious chancre appears at the inoculation site and a regional, non-tender lymphadenopathy may be palpated. Secondary syphilis, characterized by mucocutaneous lesions and high titers of blood circulating treponemes, can occur from 10 weeks to 6 months after infection. If untreated, the patient undergoes a period of latency and ultimately, the tertiary stage of syphilis, characterized by neurosyphilis, aortitis, and gummas. During the secondary, and early latent syphilis (the first 12 months of disease) mucocutaneous relapses occur and patients are usually contagious. Afterward, patients are usually not contagious, except for pregnant women. Vertical transmission via the placenta to the fetus can appear in all stages of syphilis [1,15,16,17].

TROMPs serve as antigenic molecules capable to induce immune responses via TLR-2 pathways. The predominance of treponemal proteins beneath the surface [17,18] enables these bacteria to silently disseminate, accompanied by a low systemic inflammation and symptomatology, as the innate immune system barely senses them [19]. Furthermore, the treponemal OM lacks the highly proinflammatory lipopolysaccharides (LPSs) and lipooligosaccharides (LOSs), which are usually found in Gram-negative germs [16,20,21].

As seen in the hard chancre, an initial step in the host response to *T. pallidum* is the formation of opsonic antibodies, which facilitate the internalization and degradation of treponemes through phagocytosis. The phagosomes along with the ingested treponemes further fusion with lysosomes resulting phagolysosomes. Within phagolysosomes, the treponemes are degraded and the liberated phagolysosomal lipopeptides trigger T cells recruitment. In each chancre, replicating treponemes generate an elaborate inflammatory response, consisting of macrophages, T cells, and plasma cells. Locally activated CD4+ and CD8+ T cells (with a predominance of CD4+ in early infection) produce IFN-γ. This last cytokine has a dual effect, as it stimulates the macrophages to internalize and degrade treponemes, and it provokes aggressive inflammatory cascades. Hence, the primary syphilis response is a predominant Th1 cell-based immune response. High, but inefficient titer of antibodies characterizes the secondary, humoral stage of syphilis. Secondary syphilis lesions arise after the treponemes pass across the endothelial junctions and trigger a local immune response consisting of macrophages, monocytes, and T cells [22,23,24,25,26].

### 2.2. Gonorrhea

*N. gonorrhoeae* is a species of obligate human pathogenic, facultatively intracellular, Gram-negative diplococcus bacterium that causes gonorrhea. This STI is one of the oldest recorded infections and easily spreads through unprotected sexual activity. Gonococcal primary sites of infection are the urethral mucosa, inner part of the cervix, anal canal, pharynx, and conjunctiva. Unlike *T. pallidum*, the gonococcal OMs mainly contain LOSs, LPSs, and surface hair-like proteins (pili) that facilitate movement, adherence, and genetic exchanges [1,4,7,9]. A single gonococcus can simultaneously produce up to six structurally and antigenically distinct LOSs. Because LOSs are highly expressed on the OM and are readily available as targets of the adaptive immune system, the gonococcal LOSs stand as potential vaccine candidates [27,28,29].

*N. gonorrhoeae* uses type IV pili to attach to the mucosal epithelium and other outer-membrane proteins such as opacity-associated (Opa) proteins and invasins to enable internalization. Gonococcal LOSs stimulate innate immunity and various cytokines are produced [30]. The release of pro-inflammatory cytokines such as IL-6, IL-8, IL-1β, IL-17, and IFN-γ favors polymorphonuclear leukocytes’ (PMNs) chemotaxis and infiltration to the infection area. Following recruitment to the infected tissue, PMNs phagocytose opsonized *N. gonorrhoeae*. They synthesize reactive oxygen species (ROSs) and release antimicrobial peptides (AMPs) and other molecules from their intracellular granules which induce inflammatory damage within the epithelial mucosa [31,32,33]. However, a specialized gonococcal antimicrobial efflux pump can successfully export the AMPs [34]. Hence, *N. gonorrhoeae* is able to survive and replicate inside PMNs as a result of their native resistance to antimicrobial molecules [30]. Although both the alternative and classical complement pathways may be activated during infection, the gonococcus can bind complement proteases to prevent opsonization and can successfully siaylate its LOSs as a hiding mechanism [35].

As *Treponema pallidum*, the gonococcus has the ability to modulate the adaptive immunity, as it was shown to selectively suppress Th1 and Th2 cells development and promote a Th17 response, through the induction of TGF-β. The suppression of Th1 and Th2 immune responses blocks the generation of a specific adaptive immune response, while Th17 cells recruit neutrophils [36,37,38]. Moreover, this pathogen decreases the macrophages’ and dendritic cells’ capacity to induce CD4+ T cells proliferation (Figure 1) [39,40].

### 2.3. Chlamydia trachomatis

*C. trachomatis* is a Gram-negative, coccoid or rod-shaped, strict intracellular bacterium in the genus Chlamydia, whose strains are subdivided into several serotypes. Serotypes A to C produce trachoma, serotypes D to K infect the urethral epithelia in both sexes, and endocervical epithelia of women and serotypes L1 to L3 cause lymphogranuloma venereum. Up to approximately 70% of D-K *C. trachomatis* infections are asymptomatic. Particularly in women, *C. trachomatis* infection is usually unrecognized, and not treated [1,10,41,42].

During its cellular cycle, the bacterium alternates between two morphologic types: a highly infectious, non-replicative, small form (elementary body) and a non-infectious, replicative, larger form (reticulate body) [43]. The interaction of the elementary bodies (EBs) with host cells occurs in a two-stage mechanism. The initial binding occurs through electrostatic interactions of the bacterium with heparan-sulfate containing glycosaminoglycans. This is a reversible process and is followed by a second, irreversible, binding process [44,45,46].

Upon infection, EBs are internalized inside membrane-bordered vacuoles, named intra-cytoplasmic inclusions. Within the inclusions, the EBs differentiate into the metabolically active forms, named reticulate bodies (RBs) [43]. RBs use host cytoplasmic nutrients and undergo repeated replications by binary fission throughout the middle part of the developmental cycle. The transformation of RBs back to EBs arises when the inclusion containing RBs reaches a critical volume and a depletion of nutrients and ATP occurs. The newly produced EBs are released into the extracellular milieu and another round of infection occurs [47,48,49,50]. A better understanding of the EBs metabolism has recently emerged. It appears that EBs are not metabolically inert but are capable to metabolize extracellular glucose 6-phosphate, ATP, and amino acids [51].

The host epithelial cells use surface receptors, endosomal receptors, and innate immune factors to recognize *C. trachomatis* antigens. Binding chlamydial ligands to these receptors triggers the release of pro-inflammatory cytokines and chemokines, which recruit host inflammatory cells. The phagocytosis of *C. trachomatis* triggers a specific B and T cells mediated humoral immunity. Nonetheless, this pathogen secretes potent proteases, which degrade two important transcription factors (RFX5 and USF-1) and interfere with the synthesis of major histocompatibility complex (MHC) class I and class II molecules. Chlamydia provokes a cellular potassium efflux which stimulates ROSs synthesis and eventually leads to caspase-1 activation and IL-1β and IL-18 synthesis. This inflammatory response is required for *C. trachomatis* clearance, but also promotes scarring and host damage [48,52,53,54,55]. TLRs recognize *C. trachomatis* and contribute to bacterial cleaning [56], but some innate immunity soluble factors, such as the NLRP3–ASC inflammasome [57], may promote infection by increasing fats metabolism [58].

### 2.4. Herpes Simplex Virus

HSV is a double-stranded DNA virus and a member of the Alphaherpesvirinae subfamily of the Herpesviridae family. There are two major types of HSV: herpes simplex 1 (HSV-1) and herpes simplex 2 (HSV-2). They vary both clinically and in severity degree. HSV-1 usually causes herpetic stomatitis, herpes labialis, and sight-threatening ocular herpes. Additionally, HSV-1 may produce sporadic encephalitis. While sexually transmitted HSV-2 commonly causes anogenital lesions, an increasing incidence of HSV-1 genital herpes has emerged, notably in teenagers, due to oral sexual intercourse. HSV targets two types of cells: epithelial cells and neurons. Following the oral or genital HSV infection, the virus replicates within the epidermal keratinocytes and Langerhans cells (LCs), which possess HSV receptors. HSV migrates via unmyelinated sensory nerve fibers using retrograde microtubule-associated transport (dynein and dynactin) to the neuronal soma in the dorsal root ganglion (or the sensory ganglion of the trigeminal nerve for HSV-1) [12,59,60]. 

HSV possesses several envelope glycoproteins, which have a major function in binding to host cell surface receptors and promoting HSV entry into host cells [61,62]. There are at least five important glycoproteins (gB, gC, gD, gH, gL) that were shown to enable HSV entry into the host cell. Following gB coupling with heparan-sulfate and its receptor, gD interacts with epithelial cells receptors or herpesvirus entry mediator (HVEM) on immune cells. Therefore, the recognition of HVEM by gD promotes a high viral tropism for the innate immune cells that express HVEM [63,64,65]. Furthermore, Wakeley et al. have recently shown that HVEM expression increases on CD3+ cells after trauma and plays a major role in host defense against infections in trauma patients from the ICU [66].

HSV activates the type-1 interferon (IFN) signaling pathway (with IFN-α and IFN-β production) via pattern recognition receptors (PRRs), an important family of soluble factors involved in skin innate immunity. These receptors possess specific sensors able to recognize HSV and trigger the interferon-stimulated genes (ISGs) activation. ISGs may limit HSV infection, together with a vigorous adaptive immune response majorly represented by the CD8+ T-cells production of IFN-γ (Figure 2) [67,68]. Furthermore, IFN-γ may inhibit HSV reactivation in vivo ganglia cultures [69]. Aberrations of innate immunity, such as TLR-3 mutations, are associated with herpes simplex encephalitis in children and may increase its susceptibility in adults [70].

## 3. Treatment Guidelines and Resistant Pathogens

### 3.1. Syphilis

According to the “2020 European guidelines on the management of syphilis”, benzathine penicillin G (BPG) 2.4 million units intramuscularly (IM) is the treatment of choice in early syphilis (primary, secondary, or <1 year acquired) and 2.4 million units IM weekly, for 3 weeks, in late syphilis (>1-year duration or duration unknown). Higher doses are required in neurosyphilis, while HIV patients and pregnant women should receive the same BPG regimen. A treponemicidal BPG concentration of >0.018 mg/L, for at least seven days, is warranted [71].

Procaine penicillin (IM) is a second-line therapy for early and late syphilis. During neurosyphilis treatment, probenecid should be associated with procaine penicillin. Doxycycline (PO), erythromycin (PO), azithromycin (PO), amoxicillin (PO), and ceftriaxone (IM) also represent alternative treatments to BPG [72].

In order to avoid deep painful IM BPG injection and possible allergic reactions to BPG, some clinicians may use the PO alternatives (macrolides and tetracyclines) as a first-line treatment. However, the current emergence of *T. pallidum* that displays resistance to macrolides (erythromycin, azithromycin, etc.) raises some questions regarding these PO regimens [73]. Macrolides are bacteriostatic antibiotics that bind reversibly to 23S rRNA of the 50S ribosomal subunit and inhibit protein synthesis. The A2058G and A2059G mutations were found in the treponemal 23S rRNA gene, more frequently in men who have sex with men (MSM) and HIV patients. The incidence of macrolide-resistant *T. pallidum* increased over time and its prevalence displays a geographical distribution (higher in cities and lower in remote areas) [74].

A recent study on 25 positive *T. pallidum* DNA samples from active-syphilis patients found an 88% frequency of macrolide-resistance mutations, but no resistance to tetracyclines, suggesting that macrolides should be avoided in syphilis guidelines [75]. Additionally, a 2020 study comprising 146 French patients with early syphilis found an 85% resistance to macrolides, but no point mutations for doxycycline resistance and proved the absence of genomic resistance to tetracyclines [76].

Despite the increasing prevalence of macrolide-resistant treponemes, there is no demonstrated penicillin resistance. Differentiation between relapse after an inefficient therapy and reinfection is challenging. Hence, most clinicians consider = treatment failures as a consequence of reinfection or patient-to-patient serologic variation after penicillin treatment [77]. However, literature comprises secondary syphilis relapses after IM BPG. Because IM BPG does not achieve treponemicidal concentrations in the cerebrospinal fluid (CSF), a treponemal reservoir may establish in the CSF and may serve as an origin of relapse, especially in HIV positive patients [78]. 

*T. pallidum* displays numerous chronicization mechanisms. Unlike other microorganisms, it possesses a small number of surface-exposed pathogen-associated molecular patterns (PAMPs), which could serve as potent triggering antigens for the immune response. Therefore, *T. pallidum* undergoes repeated disseminations that are hardly noticed by the innate immunity [19]. This is in line with the spread of viable treponemes during secondary and early latent syphilis despite high concentrations of specific antibodies [79]. A recent study cohort on 28 individuals [80] demonstrated this escape mechanism by measuring changes of caspase-1 and caspase-3, well-known mediators of programmed cell death. *T. pallidum* infection led to the programmed cell death of CD4+ and CD8+ T cells through both pyroptosis and apoptosis, thus hindering a robust immune response.

In addition, antigenic variations in seven variable (V) regions of the TprK gene contribute to the avoidance of developing an immune response in syphilis patients and to the chronic infection status [5]. Some studies suggest that during latency treponemes reside within hair follicles and nerves [25], thus explaining the chronic latent infection reactivation to cause tertiary syphilis in the pre-antibiotic era [26]. 

A decade ago, Babollin et al. investigated this treponemal persistence by analyzing an oligomeric protein belonging to the bacterioferritin family and produced by *T. pallidum* (TpF1). The authors found an increased serum concentration of antibodies directed against TpF1 in secondary syphilis patients, which correlated with a significantly higher mean percentage of Treg cells in infected patients when compared to healthy controls. Treg cells are a unique subpopulation of T cells that suppress immune responses and subsequently contribute to the maintenance of chronic syphilis disease [81]. TpF1 promotes vascular inflammation, angiogenesis, and cardiovascular complications in secondary and tertiary syphilis patients [82].

### 3.2. Gonorrhea and Chlamydia

According to the “2020 European guideline for the diagnosis and treatment of gonorrhea in adults”, dual therapy comprises a single-dose ceftriaxone 1 g IM and azithromycin 2 g PO. This therapy targets both intracellular and extracellular *N. gonorrhoeae* and provides excellent cure rates. Ceftriaxone IM alone may be considered when azithromycin is not available, or there is clear evidence of absent ceftriaxone resistance. Additionally, doxycycline can be used as an alternative to azithromycin in combination with ceftriaxone. Other antibiotics such as spectinomycin, ciprofloxacin, and gentamicin can be considered in various settings [83].

The first *N. gonorrhoeae* FC428 clone resistant to ceftriaxone appeared in Japan in 2015, presumably caused by the mutation of a mosaic allele from an oropharyngeal Neisseria [84]. This aggressive clone has lately spread globally, while the *N. gonorrhoeae* resistance rate saw a worrying ascension. A Chinese study revealed an increase of the ceftriaxone-decreased-susceptibility from 2.05% (2016) to 16.18% (2019) [85]. Therefore, this global rapid ascension of the FC428 *N. gonorrhoeae* jeopardizes the current treatment guidelines and may announce a hereafter global AMR to ceftriaxone and cefixime in gonorrhea patients [86]. 

Additionally, in 2022, an increasing prevalence of azithromycin-resistant gonococcus was described in Canada and northern Spain. Most cases are associated with the clonal spread of the ST-12302 genogroup, as ST-12302 gonococcus was lately acknowledged as an epidemic clone [87,88].

In 2012 the CDC ranked drug-resistant *N. gonorrhoeae* as a “superbug” [89] and in 2017 the World Health Organization (WHO) classified it as a “High Priority” pathogen in “WHO Global Priority List of Antibiotic-Resistant Bacteria to Guide Research, Discovery, and Development of New antibiotics” [90]. 

Because co-infection with Chlamydia is frequent in men who have sex with men (MSM) and young patients with gonorrhea, the treatment guidelines for gonorrhea usually cover Chlamydia co-infection [83]. The 2015 treatment guidelines for uncomplicated urogenital *C. trachomatis* infections recommend the 7-days therapy with doxycycline (100 mg PO twice a day) or azithromycin 1 g orally once as a first-line treatment. Second or third-line treatment regimens comprise erythromycin, levofloxacin, ofloxacin, josamycin, or rifampicin [91].

*C. trachomatis* reinfection in treated women may arise as a consequence of the persistent gastrointestinal infection. Therefore, women who are free of genital *C. trachomatis* infection remain at risk for gastrointestinal autoinoculation. Although there is no significant difference in azithromycin concentration between the cecum and the cervix, the regular azithromycin dosage is inefficient against gastrointestinal infection. Hence, azithromycin has a lower efficacy against chlamydial gastrointestinal infection [92,93]. 

Furthermore, some authors suggest that single-dose azithromycin has a lower efficacy against rectal chlamydiosis when compared to the 7-days doxycycline regimen and suggest azithromycin effectiveness as first-line therapy should be revisited [94,95]. Chlamydia trachomatis may develop AMR to macrolides via 23S rRNA mutations, to tetracyclines via mutations in the tet(M) gene, and to fluoroquinolones via mutations in the gyrA, parC, and ygeD genes. However, the majority of the mutations are found in the 23S rRNA gene and promote AMR to macrolides [96]. 

A 2020 study comprising 92 patients with genital chlamydiosis found higher detection rates of 23S rRNA and tet(M) gene mutations in the treatment failure sample and comparable minimum inhibitory concentrations (MICs) between the two samples. The authors suggest that detection of AMR genes could better explain the high treatment failure rates than the MICs and endorse the necessity of genetic AMR testing [97].

Genital *N. gonorrhoeae* and *C. trachomatis* infections may transform into chronic diseases and produce comparable injuries to the pelvic organs, both in men and women. In approximately 10% of the male cases, the *N. gonorrhoeae* infection is asymptomatic. If untreated, it can produce prostatitis, orchitis, or chronic epididymitis [98]. Polymerase chain reaction (PCR) can be a useful tool for the diagnosis of chronic gonorrhea prostatitis. Unlike HSV-2 infection, a study on 243 paraffin-embedded prostate tissues obtained from patients with hyperplasia and prostate cancer found no statistically significant correlation between prostate cancer and *N. gonorrhoeae* [99]. Contrarily, a meta-analysis comprising 9965 prostate cancer patients demonstrated a relation between *N. gonorrhoeae* prostatitis and prostate cancer [100]. Young males infected with *N. gonorrhoeae* tend to have a higher titer of prostate-specific antigen (PSA), sustaining the evidence of prostate infection and inflammation [101].

Both the gonococcus and *C. trachomatis* may produce acute and chronic bacterial prostatitis in sexually active individuals. In vitro prostatic epithelial cells respond to *C. trachomatis* infection by producing an inflammatory response. Hence, high titers of IL-8 may be detected in the semen of these patients. Furthermore, an association of *C. trachomatis* prostatitis and decreased sperm concentration and sperm cells viability, motility, and normal morphology was reported [102]. 

Both acute and chronic bacterial prostatitis respond to antibiotic therapy (the latter usually requires longer treatment durations). If bacterial prostatitis is not completely treated, the prostate may become a bacterial reservoir. However, the most common prostatitis (more than 90% of prostatitis cases) is chronic non-bacterial prostatitis (CNBP). Several studies have demonstrated *C. trachomatis* potential to trigger CNBP, even in the context of paraclinical confirmed *C. trachomatis* eradication. Prostatic chronic inflammation usually arises in genetically susceptible individuals following the initial recognition of *C. trachomatis* antigens by TLR-2 and TLR-4. Furthermore, an autoimmune prostatic process can be initiated. Blood serology and urine PCR test may distinguish between bacterial and non-bacterial prostatitis [99,103]. 

*N. gonorrhoeae* and *C. trachomatis* are among the most commonly acknowledged pelvic inflammatory disease (PID) pathogens (one quarter to one-third of PID cases) [104]. Tubo-ovarian abscess, hydrosalpinx, pyosalpinx, and oophoritis are common PID clinical features [105].

*N. gonorrhoeae* and *C. trachomatis* are responsible for tubal-factor infertility (TFI), which accounts for 30% of female fertility issues. While *N. gonorrhoeae* can impair ciliated cells’ function within the fallopian tubes, *C. trachomatis* can infect both ciliated and non-ciliated cells, but the ciliary function is not affected. Additionally, *C. trachomatis* determines disruption of intercellular junctions and loss of the microvilli [106,107]. The heat shock protein 60 (hsp60) produced by *C. trachomatis* induces tubal inflammation, resulting in scarring and tubal blockage, with tubal infertility. Hsp60 combined with chlamydial protease-like activity factor can predict TFI [108,109,110]. Regardless of tubal patency, Chlamydia antibodies associate increased ectopic pregnancy risk and lower chances of pregnancy [111].

### 3.3. Herpes Simplex

For both first-episode and recurrent genital herpes, the treatment of choice consists of one of the well-known oral antiviral guanosine analogs (acyclovir, famciclovir, valaciclovir), with various duration and dosages. The usual first-episode genital herpes regimen comprises acyclovir 400 mg PO three times a day, for five to ten days. The immunocompromised patients may develop acyclovir-resistant HSV infections and often require different management, especially during relapses or unretractable herpes infection [112].

Guanosine analogs require phosphorylation by HSV-thymidine kinase (TK) to its active form. However, strains of HSV manage to acquire resistance to guanosine analogs by eliminating TK or decreasing its affinity for the guanosine analogs. In HIV-positive patients, impaired cell-mediated immunity allows these resistant strains to replicate and delays the clearance. Foscarnet and cidofovir represent the first-line treatment in acyclovir-resistant HSV/TK deficient strains since these intravenous antivirals do not need viral TK for activation, with the cost of potential nephrotoxicity [112,113].

In HIV-negative immunosuppressed patients with refractory acyclovir-resistant HSV2, pritelivir, an oral helicase-primase inhibitor, was successfully used, paving the road towards new treatments for recalcitrant infection [114]. These clinical results are in line with some recent preclinical, mouse model findings, that showed superior effectiveness of helicase primase inhibitors when compared to the nucleoside analogs [115].

Imiquimod is an imidazoquinoline amine that upregulates cystatin A synthesis in HSV infected nonimmune cells and triggers an efficient IFN-independent anti-HSV response. Additionally, imiquimod is a potent agonist for TLR-7 that triggers strong immune responses via recognition of the HSV in the infected cells [57]. Topical imiquimod 5% was successfully used in some HIV patients with acyclovir-resistant anogenital HSV infection [116]. Furthermore, vegetative chronic genital herpes in HIV patients may be responsive to Imiquimod 5% [117].

Chronicization mechanisms in HSVs arise as a direct consequence of their life cycle. HSV-1 and HSV-2 infect epithelial cells at the beginning (the lytic stage of infection) and subsequently target sensory neurons (a life-long, latent infection stage). During the lytic stage, painful, highly contagious sores appear. However, during the latent stage of infection, HSV is dormant within neurons, no infectious virions are produced, and the host is not contagious. When HSVs undergo the dormant stage within the neuronal karyon, its input linear DNA circularizes by an end-to-end ligation and may provide the replication template [118,119].

Autophagy represents an important host defense mechanism to get rid of pathogens. HSVs inhibit autophagy by binding to Beclin 1, which prevents autophagophore formation. Moreover, Waisner and Kalamovky showed that herpetic infected cell protein 0 (ICP0) downregulates two major autophagy adaptor proteins: sequestosome 1 (p62/SQSTM1) and optineurin (OPTN). These proteins regulate innate immunity and inflammation and have an antiviral function [120]. ICP0 additionally limits the anti-viral type-I IFN mediated immune responses [121].

The latent herpetic genome is associated with repressive heterochromatin, a nucleosomal structure, and cellular histones. Chromatin organization is of critical importance for the maintenance of the HSV dormancy stage. Local trauma, emotional stress, fever, other infections, sun exposure, or hormonal disturbances may initiate the lytic stage. HSV reactivation requires both triggers that may induce the synthesis of HSV lytic genes and a decrease in the effectiveness of the immune response (Figure 3) [122]. Following viral genes synthesis and reassembly, HSV undergoes an anterograde transport towards the axon tips. At this point, the virus can spread across cell junctions and epithelial cells that participate in neuronal synapses and the host becomes contagious. A well-studied anterograde axonal carrier for HSV along microtubules is kinesin-1 [123]. Recent studies suggest that HSVs incorporate kinesins proteins and use them to produce” motorized viral particles” [124].

## 4. Novel Drugs and Nano Molecules

### 4.1. Syphilis

Despite more than a century of sustained struggle, exhaustive research of *Treponema pallidum* has been impeded by the inability to culture the bacterium continuously in vitro. In 2018, Edmondson et al. managed to culture treponemes in vitro, obtaining a long-term multiplication. They have used a modified medium with a rabbit epithelial cell coincubation system which managed to maintain six months of continuous treponemes growth [125]. Their discovery facilitated future in vitro studies and novel antibiotics were studied on these cultures.

A 2021 preclinical study [126] revealed an in vitro linezolid bactericidal activity against *T. pallidum* at concentrations of a minimum of 0.5 µg/mL. In vivo efficacy was demonstrated on fifteen rabbits infected intradermally with *T. pallidum*. Oral linezolid showed healing of early cutaneous lesions at a comparable time to BPG. *T. pallidum* absence at dark-field microscopy was noted after three days of therapy. Contrarily, moxifloxacin and clofazimine failed to inhibit *T. pallidum* growth in vitro and to cure syphilis in the rabbit models.

D-alanyl-D-alanine ligase inhibitors may represent the key to some drug-resistant germs, including *T. pallidum*. This enzyme is involved in the biosynthesis of peptidoglycans, which are essential in maintaining the integrity of the bacterial cell wall. Specific natural molecules and kinase inhibitors have been found to work as D-alanyl-D-alanine ligase inhibitors, but some may find a difficult entry through the thick bacterial wall [127]. More than two decades ago, Qing et al. found that a modified diterpenoid quinone (salvicine) exhibits strong cytotoxic activity against various human tumor cell lines [128]. Later on, it was shown that salvicine not only inhibits topoisomerase-II activity and tumor growth but promotes ROSs synthesis and possesses an anti-multidrug-resistant (MDR) activity [129]. In 2015, Dwivedi et al. showed that 126 proteins are essential for *T. pallidum* viability and pathogenicity, and 106 of these are connected to vital metabolic pathways. Six pathways, including the D-alanyl-D-alanine ligase pathway, were found unique for *T. pallidum*. Targeting D-alanyl-D-alanine ligase with salvicine represented the most potent inhibition of the upper pathways with plant-derived terpenoids and opened the door towards the use of D-alanyl-D-alanine ligase inhibitors in syphilis [130].

### 4.2. Gonorrhea

Because *N. gonorrhoeae* may be resistant to first-line therapy with ceftriaxone, alternative treatments are quite urgent.

A randomized, controlled, double-blind, non-inferiority trial found earlier this year that single-dose 1000 mg ertapenem (a broad-spectrum carbapenem) is not inferior to single-dose 500 mg ceftriaxone [131].

The only currently approved aminoacyl tRNA synthetases (aaRS) inhibitor is mupirocin. Cephalosporin-resistant gonococcus was found sensitive to a PEGylated nano-liposomal formulation of this isoleucyl tRNA synthetase inhibitor (nano-mupirocin). This novel drug retains full potency against other bacteria, such as resistant Staphylococcus aureus and Enterococcus faecium [132,133].

In 2018, a spiropyrimidinetrione that inhibits bacterial type-II topoisomerases (zoliflodacin) was found safe after single doses in healthy recruits [134]. Three years later, an in vitro study of 1209 consecutive clinical *N. gonorrhoeae* isolates did not reveal any cross-resistance to ciprofloxacin, azithromycin, cefixime, and ceftriaxone [135]. Patients should take at least 2 g of a single oral dose treatment of zoliflodacin to obtain effective *N. gonorrhoeae* suppression [136]. Additionally, studies revealed that the conventional therapeutic dose of 2 g does not induce cardiac arrhythmias and is well tolerated [137]. Although zoliflodacin maintains an excellent in vitro activity against clinical *N. gonorrhoeae* isolates, the rising potential to develop AMR led to further antibiotics research [138]. A 2021 study found that a single dose of MBX-4132, an acylaminooxadiazole that selectively inhibits ribosomal trans-translation, successfully cleared MDR *N. gonorrhoeae* infection in mice [139]. This molecule pioneered trans-translation inhibitors use in MDR *N. gonorrhoeae*.

### 4.3. Chlamydia

The emergence of resistant *C. trachomatis* has determined worldwide scientists to search for novel drugs against these strains. Worryingly, recent evidence suggests the regulation of mitochondria-mediated host cell apoptosis as a survival in vivo mechanism [140]. Corallopyronin A is an antibacterial molecule synthesized by *Corallococcus coralloides*. It binds the bacterial DNA-dependent RNA polymerase and successfully inhibits both the wild type and the rifampicin-resistant *C. trachomatis*. This molecule was also efficient against *C. trachomatis* in ex vivo human fallopian tube model [141,142].

Yang et al. found in 2019 a nanoparticle formulation able to impair one of *C. trachomatis* surface binding proteins. This nanoparticle (PDGFR-β siRNA-PEI-PLGA-PEG NP) significantly caused autophagy in human vaginal epithelial cells and decreased a *C. trachomatis* surface protein (PDGFR-β) gene expression [143].

Other attempts to decrease the use of the antibiotic in genital chlamydiosis have arisen, as an attempt to protect the commensal flora. Recently, a 2-pyridone amide was discovered to inhibit transcription of crucial genes in progeny EBs, which became unable to differentiate into the RBs [144].

### 4.4. Herpes Simplex

Recently, a peptide isolated from a marine bacterium (*Micromonospora* sp.), was shown to display a specific inhibitory effect against HSV-2. Transmission electron microscopy revealed that this peptide diminishes the viral spread by containing the herpetic virions escape from the infected cells and suppresses the herpetic cytopathic effect. Furthermore, it is potent against acyclovir-resistant HSV-2 [145].

At least nineteen antiviral active molecules derived from natural sources, such as harmine, emodin, casuarinin, and oxyresveratrol, could represent promising alternatives for treating HSV-2 infections. Additionally, some of these natural molecules may be effective in acyclovir-resistant infections [146].

Nano-based materials, such as metal nanoparticles, polymeric nanoparticles, micelles, and dendrimers, were extensively studied in the last decade as herpetic inhibitors. These nano molecules bind the surface glycoproteins, such as gC and g120, and inhibit the viral entrance within cells [147].

Vaginal treatment with topical acyclovir is frequently inefficient due to its low bioavailability in the vaginal environment. Nano molecules used to carry acyclovir were extensively studied in order to increase the vaginal acyclovir potency. In a recent study, acyclovir was complexed with sulfobutyl-ether-β-cyclodextrin and then integrated into the nanodroplet chitosan shell. This nanocarrier exhibited enhanced antiviral activity in HSV-2 infected cell cultures when compared to the free acyclovir [148]. Additionally, some other studies revealed that zinc oxide nanoparticles and acyclovir monophosphate-loaded mucus-penetrating nanoparticles provide efficacious protection against HSV-2 infection. Acyclovir nanoparticles exhibit a maintained release, low-grade toxicity, better penetration, and diffusion into the cervical tissue (Figure 4) [149].

## 5. Conclusions and Future Research Perspectives

*Treponema pallidum* remains a peculiar and still not completely deciphered pathogen even after a century of investigations. Although it is generally acknowledged that having syphilis once does not grant protection from reinfection, clear scientific evidence regarding reinfection mechanisms has not emerged yet. Prostatic and gastrointestinal STIs reservoirs represent an important therapeutic challenge and conventional drugs may be inefficient in preventing reinfections or sexual transmission. Tubal infertility associated with asymptomatic STIs has become a global health issue.

Given the early successes in developing *Treponema pallidum* cultures, high attention on shaping new molecules to be studied in preclinical studies has appeared. Linezolid and D-alanyl-D-alanine ligase inhibitors serve as potential treponemicidal drugs and MDR-*T. pallidum* inhibitors, respectively. However, the new emerging ceftriaxone-resistant *N. gonorrhoeae* represents a more urgent threat than resistant *T. pallidum*. Mupirocin and zoliflodacin are currently used against this dangerous strain, and novel ribosomal trans-translation inhibitors were lately proposed. Imiquimod may represent a successful treatment against recalcitrant HSV, especially in immunocompromised patients.

Nano-based materials have become of great interest, notably in *C. trachomatis* and HSV vaginal infections, as they may increase local absorption. Molecules derived from natural products and bacteria-produced medications shed light on non-antibiotic therapies in genital chlamydiosis and herpetic infections. During the last ten years, important steps were made to mobilize attention towards these AMR infections. A switch towards translational medicine is warranted in this matter. Further discoveries in the field of nanomaterials will increase medications’ potency in the battle with these germs.

## Figures and Tables

**Figure 1 ijms-23-03550-f001:**
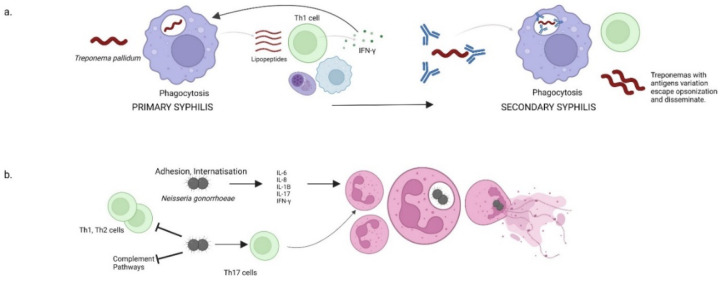
Host responses in syphilis and gonorrhea. (**a**) A dominant Th1 response in primary syphilis promotes IFN-γ production. Opsonized treponemas in secondary syphilis undergo phagocytosis and resistant treponemas disseminate and produce skin and mucosal lesions. (**b**) *Neisseria gonorrhoeae* inhibits complement pathways and Th1 and Th2 cells but stimulates Th17 cells which promote neutrophil infiltration. Some gonococci may escape phagocytosis and invade pelvic organs.

**Figure 2 ijms-23-03550-f002:**
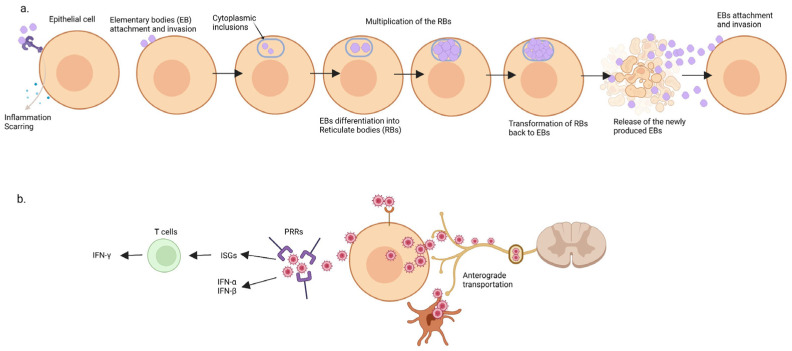
Host interaction in Chlamydia and herpes simplex. (**a**) Chlamydia life cycles alternate between the elementary bodies and the reticulate bodies. Chlamydiosis inflammation may resolve with local scarring. (**b**) Herpes simplex virus binds pattern recognition receptors (PRRs) and stimulates IFN (α, β, γ) production. Following proliferation within the epidermal keratinocytes and Langerhans cells, the herpes virus undergoes anterograde transportation and enters the dormant stage inside the dorsal root ganglia. ISG = interferon-stimulated genes; PRR = pattern recognition receptor.

**Figure 3 ijms-23-03550-f003:**
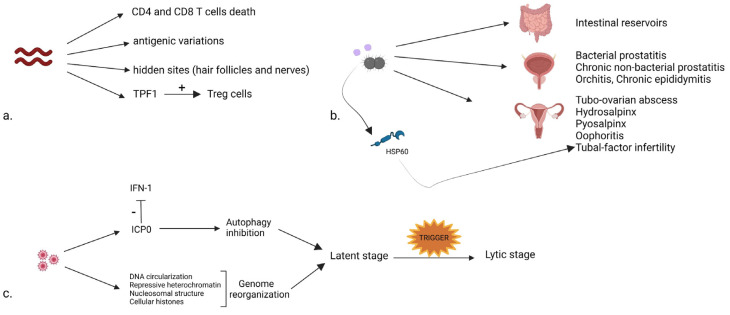
Chronicization mechanisms. Syphilis (**a**), *Neisseria gonorrhoeae* and Chlamydia (**b**), and Herpes Simplex (**c**) display complex immune escape mechanisms within hair follicles, nerves, pelvic organs, and intestines. TPF1 = treponemal protein belonging to the bacteriominiferritin family; HSP60 = heat shock protein 60; ICP0 = infected cell protein 0.

**Figure 4 ijms-23-03550-f004:**
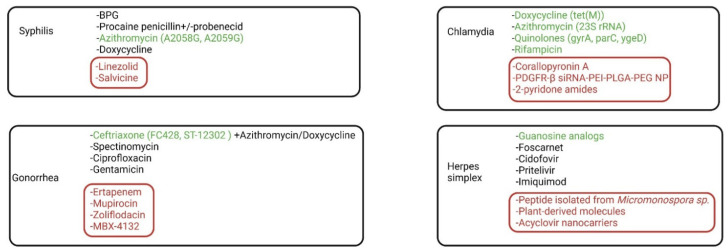
Current no demonstrated drug-resistance (black color), demonstrated drug-resistance (green color), and novel molecules (red color). Plant-derived drugs, nanoparticles, D-alanyl-D-alanine inhibitors, topoisomerase inhibitors, and novel antibiotics provide excellent options against worldwide emerging drug-resistant sexually transmitted germs.

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
