# Peer review of "Fascinating Molecular and Immune Escape Mechanisms in the Treatment of STIs (Syphilis, Gonorrhea, Chlamydia, and Herpes Simplex)"

_ijms, 2022, doi:10.3390/ijms23073550_

Round 1

Reviewer 1 Report

This is a good review of the current state of STI pathophysiology, management, antimicrobial resistance and prospective new therapies.

There are some of corrections required:

Line 48 : "promote immunity escaping" please correct the grammar

Line 66: How do you know that T. pallidum is gram negative when it cannot be viewed by light microscopy and electron microscopy and darkfield microscopy have no staining?

Line 68: If Tp is deposited onto intact skin and mucosa, there will be no infection. This organism is inoculated into skin and mucosa via micro-abrasions. Please correct this.

Line 71/72: "Up to ten weeks after the primary lesion resolution,.." this implies that secondary syphilis can occur from infection up to 10 weeks after the primary lesion resolves. Secondary syphilis can occur from 10 weeks to 6 months after infection.

Line 73: "high titers of blood circulating Tp, arises." This is likely true, however, it is not born out by PCR which has a very low sensitivity in blood, in primary and secondary syphilis.

line 75: "During the secondary, and early latent syphilis (the first four years)" early latent disease is during the first 12 months of infection; late latent disease is after the first 12 months of disease.

Line 88: "Subsequently to Tps’ ingestion in the phagosomes" please correct the grammar

Line 98: "Tp traverses the junctions between endothelial junctions" please correct

Line 105: "Ngo OM majorly contains LOSs and LPSs"  please correct the highlighted typo and correct the grammar.

Lines 108-109: This will be difficult since one of the mechanisms of host immune evasion of this organism is masking the LOS antigens via sialylation, as mentioned in line 124.

line 125: "Similar to Tp, Ngo has the.." please correct the highlighted typo

Section 2.3 on Chlamydia trachomatis does not mention any host immune response as in the previous 2 sections and organisms. To be consistent, this should be included.

Line 241; "therapy and reinfection imposes to consider most treatment failures as a consequence of reinfection or patient-to-patient serologic variation following penicillin treatment "  please correct the grammar

Line 243: "the literature comprises relapses of secondary syphilis after IM BPG" please correct

Line 263: 'bacteriominiferritin" should this be bacterial mini-ferritin?

Line 269: "is capable to promote" please correct the grammar

Line 353: "Ct determines the disruption of intercellular junctions and microvilli loss" there is likely a word missing from this sentence.

Line 357-358: "Ct antibodies associate with increased ectopic pregnancy risk and a lower likelihood of pregnancy, even in the presence of tubal patency" please correct the grammar

Line 411: " produce motorized viral particles” [124]. " please correct

Line 426: "revealed an in vitro linezolid bactericidal against Tp activity at concentrations of a minimum" please correct the grammar

Line 428: " Orally linezolid showed healing"  please correct the grammar

Line 439: tumor growth

Reviewer 2 Report

Scurto et al have reviewed an important aspect of the major causes of sexually transmitted infections, their immune-activating reactions as well as treatment strategies. Overall, the paper is well written and contributes to the scientific community.

I have some minor comments:

  1. Please use N. gonorrhea and not Ng, T. pallidum and not Tp, and C. trachomatis and not Ct.
  2. There are several typos in Figure 2, for instance, cel instead of cell, intro instead of into.
  3. English language of the whole manuscript needs to be edited. Some examples: lines 12-13, lines 106-107.
